# A Review on Smartphone Keystroke Dynamics as a Digital Biomarker for Understanding Neurocognitive Functioning

**DOI:** 10.3390/brainsci13060959

**Published:** 2023-06-16

**Authors:** Theresa M. Nguyen, Alex D. Leow, Olusola Ajilore

**Affiliations:** 1Department of Psychiatry, University of Illinois at Chicago, Chicago, IL 60612, USA; 2Department of Biomedical Engineering, University of Illinois at Chicago, Chicago, IL 60607, USA; 3Department of Computer Science, University of Illinois at Chicago, Chicago, IL 60607, USA

**Keywords:** smartphone, digital technologies, passive data collection, cognition, keystroke dynamics

## Abstract

Can digital technologies provide a passive unobtrusive means to observe and study cognition outside of the laboratory? Previously, cognitive assessments and monitoring were conducted in a laboratory or clinical setting, allowing for a cross-sectional glimpse of cognitive states. In the last decade, researchers have been utilizing technological advances and devices to explore ways of assessing cognition in the real world. We propose that the virtual keyboard of smartphones, an increasingly ubiquitous digital device, can provide the ideal conduit for passive data collection to study cognition. Passive data collection occurs without the active engagement of a participant and allows for near-continuous, objective data collection. Most importantly, this data collection can occur in the real world, capturing authentic datapoints. This method of data collection and its analyses provide a more comprehensive and potentially more suitable insight into cognitive states, as intra-individual cognitive fluctuations over time have shown to be an early manifestation of cognitive decline. We review different ways passive data, centered around keystroke dynamics, collected from smartphones, have been used to assess and evaluate cognition. We also discuss gaps in the literature where future directions of utilizing passive data can continue to provide inferences into cognition and elaborate on the importance of digital data privacy and consent.

## 1. Introduction

In 2021, 97% of Americans owned a phone, with 85% of them owning a smartphone [1]. In the developing world, 45% of people have smartphones, with the number growing daily [1]. Until recently, cognitive testing has been conducted within a laboratory or clinical setting, but with the advent of technological advances, smartphones and other wearable technologies have provided new tools for remote cognitive testing in the real world. As more smartphones are used and become truly ubiquitous devices worldwide, the research potential for longitudinal active and passive data collection increases proportionally. Active data collection is when participants are prompted to perform a task, whereas passive data are collected unobtrusively with participants being unaware of the data collection; definitions and examples for both active and passive data are given in Table 1. Ecological momentary assessments (EMAs) are an example of active data collection and have gained considerable traction within the last decade with the administration of EMAs via smartphones. Participants receive notifications on their smartphones at specified times during the day to complete surveys and other tasks. EMAs offer researchers a method to assess participants (e.g., their thoughts and feelings, motor/cognitive/mood assessments) in real time and in their natural environment, which decreases the probability of recall bias [2]. However, active participation needed for EMAs may gradually yield less data over time as participants eventually stop using or have low participation rates for application-based activities and interventions [3]. To augment research capabilities of smartphones, researchers have turned to passive data collection, which increases the amount of information acquired while decreasing burdens on participants. Passive data collection is when data from smartphone sensors (e.g., GPS, accelerometer, keystroke dynamics) are acquired from participants unobtrusively, yet consensually, and thus goes unnoticed by participants. Current smartphone sensors can include precision dual-frequency global positioning system (GPS), digital compass, iBeacon microlocation, barometer, high dynamic range gyro, high g accelerometer, proximity sensors, dual ambient light sensors, and temperature sensors [4]. Figure 1 depicts the types of passive data that can be collected via a smartphone and its sensors. Using smartphone applications that can passively register activity in the background during usage but not record the content itself provides researchers with unparalleled access to data while still allowing for privacy. Additionally, it allows for objective data collection, as some self-reported measures have been shown to be less accurate when compared to passively collected data [5]. Passive data, because of its unobtrusive, longitudinal, objective and near-continuous collection, can provide researchers with insights into cognition and cognitive fluctuations outside of a laboratory setting and can reveal potential biomarkers for neuropsychiatric disorders. Moreover, keystroke dynamics and other passive data may provide better insights into cognition as cognitive tests in a laboratory setting provide limited insight—“snapshots” per se of cognition—compared to a more complete picture outside of a laboratory setting. For example, some cognitive tests focus on speed and reaction time, which may not realistically reflect how different cognitive processes relate to or modulate one another in real life. During these types of tests, participants are placed in a controlled environment devoid of usual day-to-day distractions, while at the same time, are cognizant of being observed and have the additional stress of needing to perform well [6,7]. In addition, patients who participate in clinical research may have required periodic testing to monitor disease course, or they may wish to participate in research but are hindered by the number of visits. Using smartphones to monitor disease progression and conduct research would decrease this burden. Passive data collection via smartphones provides a way to circumvent this barrier to long-term participation and makes research more accessible to a greater number of participants. 

Within the last decade, researchers have used passive data collection via smartphones to investigate cognition. Preliminary results have shown that passive data collection can possibly be used in lieu of laboratory-based neuropsychology assessments [8]. Currently, bedside clinical screening tools for cognitive assessment may include the mini mental state examination (MMSE) [9], the abbreviated mental test [10], the mental status questionnaire [11], the short portable mental status questionnaire [12], and the Montreal cognitive assessment [13]. These rapid assessments are meant to be quick, cost-effective evaluations of cognition, but can be limited in their specificity. These clinical screening tools would then lead to additional in-depth neuropsychological assessments which require in-person assessments and yield only a cross-sectional view of cognition at the time of assessment. Smartphones would allow for not only accessible, longitudinal remote monitoring and assessments of intra-individual cognitive fluctuations, but also passive unobtrusive data collection, where participants are unaware that objective data are being collected. 

One of the primary ways users actively interact with their smartphone (instead of merely passively browsing) is through keypresses and related keyboard dynamics (simply referred to as keystroke dynamics hereafter) that are passively collected via a modern smartphone’s virtual keyboard. Keystroke dynamics refer to keypress-related metadata (e.g., general category of the keypress, corresponding timing information of key down press and release time, incidences of autocorrect, etc.) on the smartphone keyboard but not the actual text. Intuitively, typing on a smartphone keyboard utilizes multiple cognitive domains. Articulating thoughts by typing on a smartphone keyboard requires awareness of both psychomotor and visuospatial processes [14]. Given the necessary cognitive and motor processes that must be engaged to type efficiently on a smartphone keyboard, it is plausible that cognitive deficits or dysfunction could be detectable via keystroke dynamics. In addition, fine, individualized motor movements can be sampled by triggering the accelerometer and/or gyroscope, thus opening up possibilities of detecting any subtle motor anomalies before any clinically diagnosable symptoms arise [15] and may provide important digital biomarkers to serve as advanced warnings of brain dysfunction. Moreover, quantitatively characterizing cognitive processes is particularly important given how their dysfunction is the basis of a plethora of disorders. With smartphones being increasingly used in data collection and research, we sought to summarize current research using keystroke dynamics to elucidate processes within the cognitive domain, as defined by the Research Domain Criteria (RDoC), as well as discuss future directions and the ethicality of using passive data collected from smartphones and other wearable technologies. 

### 1.1. Defining Cognition and Its Domains

There are a multitude of ways to consider, study, and understand cognition. Relevant to this review, the National Institute of Mental Health created a research framework to collectively understand neuropsychiatric disorders: the Research Domain Criteria (RDoC) [16]. RDoC introduces one way of understanding cognition, where cognition consists of six domains: attention, working memory, declarative memory, language, cognitive control, and perception [17]. Attention refers to “a range of processes that regulate access to capacity-limited systems, such as awareness, higher perceptual processes, and motor action” [16], and perception is defined as the intake of sensory information which can then guide action [16]. Articulating thoughts by typing on a smartphone keyboard requires awareness of both psychomotor and visuospatial processes [14]. Neurodegenerative disorders have been shown to have both non-spatial and spatial deficits of visual attention and perception [18,19], and neuropsychiatric disorders have been shown to have some perception deficits [20,21] and attentional deficits in general [22]. Additionally, language is equally important to type coherently on a smartphone keyboard. Language is defined as “a system of shared symbolic representations of the world, the self and abstract concepts that supports thought and communication” [16]. The ability to communicate and to express desires and thoughts is vital for both physical and mental well-being. Given the complex networks within the brain that facilitate communication, language is an important marker of cognitive functioning, with semantic dysfunction manifesting earlier in certain neurodegenerative disorders than other symptoms [23,24,25]. Cognitive control is also important to cognition as it regulates the operation of cognitive and emotional systems to accomplish goals [16]. Neurodegenerative disorders can be distinguished by a progressive decline in gross and fine motor control and cognitive control, with evidence indicating disturbances in cognitive processes associating with motor deficits [26], which may be evident via changes in keystroke dynamics on smartphones. Lastly, declarative memory refers to the obtainment, consolidation, and retrieval of facts and events [16], while working memory is defined as “the active maintenance and flexible updating of goal/task relevant information (items, goals, strategies, etc.) in a form that has limited capacity and resists interference” [16]. Working memory is required when multitasking to keep certain tasks in mind (e.g., switching between smartphone applications for multiple tasks). Deficits of both types of memory have been shown in neuropsychiatric and neurodegenerative disorders [27,28]. Executing a series of keypresses efficiently and without errors may utilize all cognitive domains. For example, someone navigating public transit on the way to work while simultaneously trying to type long complex sentences with difficult-to-spell words would likely employ all six cognitive domains to accomplish their task. As motor and cognitive deficits increase in certain disorders, smartphone measurements can potentially monitor symptoms and provide information regarding the state of cognitive domains. 

### 1.2. Intraindividual Variability 

Intraindividual variability (IIV) refers to fluctuations in cognitive performance for tasks repeated over time. IIV consists of two categories: (1) inconsistency, the variability of cognitive performance in a single task over a short period of time, and (2) dispersion, the variability of cognitive performance across different tasks over time [29]. Reaction time, finger tapping, and memory capacity can be measured from these assessments and compared longitudinally. Previous research has shown possible links between IIV and neurological dysfunction, with IIV being a potential earlier marker for the initial cognitive changes associated with the onset of neurodegenerative diseases, such as Alzheimer’s disease [30,31], multiple sclerosis [32,33,34,35,36], as well as mild cognitive impairment [37,38,39]. IIV is also more sensitive during prodromal stages of neurodegenerative disorders and is a strong predictor of progressive cognitive decline [40,41,42]. Previously, smartphones have been used to measure cognition and IIV in participants with neurodegenerative disorders using EMAs [43,44,45], with some participants being asked up to six times a day to complete assessments and with varying adherence rates. By using keystroke dynamics to measure IIV and infer cognitive states, researchers would lessen any burdens which active data collection would place on participants and be able to obtain data from all participants, given the nature of passive data collection. 

## 2. Methods 

### 2.1. Search Strategy, Eligibility Criteria, and Selection Process

We conducted a number of searches to identify studies that used keystroke dynamics and other passive data types to assess neurocognitive functioning following the Preferred Reporting Items for System Reviews and Meta-Analyses guided (PRISMA) guidelines, with a flow diagram depicted in Figure 2. We initialized the search using these terms: “passive data” AND “cognition” AND/OR “smartphone” AND/OR “keystroke dynamics” AND/OR “keyboard dynamics” AND/OR “accelerometer” on PubMed and Embase with no filters. Google Scholar was additionally utilized. Searches were conducted until April 2023. Only studies written in English were included in this review. Studies were only included if they fulfilled the following criteria: (1) collected and analyzed passive data, specifically keystroke dynamics; (2) used smartphones to collect data; and (3) assessed cognitive functioning. Studies were then excluded if they: (1) did not assess cognition as defined by RDoC; (2) evaluated cognition but did not employ keystroke dynamics; and (3) used non-smartphones to collect keyboard dynamics. We analyzed all search results systematically by title, abstract, and keywords initially for relevancy and eligibility, followed by a full-text evaluation.

### 2.2. Data Extraction 

The search yielded ten studies that met the inclusion criteria, which are summarized in Table 2. The following data were extracted from these studies into Microsoft Excel and Word tables: types of active and passive data collected, digital technologies used, overall findings, and types of analyses conducted. 

## 3. Keystroke Dynamics and Affected Cognitive Domains in Neurodegenerative Disorders 

### 3.1. Alzheimer’s Disease and Mild Cognitive Impairment

Alzheimer’s disease (AD) is a progressive neurodegenerative disease characterized by the gradual loss of motor function and cognitive facilities [56]. Studies have shown that language and speech can manifest as part of the early signs of mild cognitive impairment (MCI) and other prodromal stages of AD while correlating with declines in episodic and semantic memory [23,24]. These studies have also indicated that there may be a preclinical AD stage where cognitive, behavioral, sensory, and motor changes can possibly precede clinical manifestations of AD by years [24]. Researchers have examined how language characteristics change in participants with AD and found that AD can already influence temporal characteristics of spontaneous speech (i.e., increased hesitations) and in reading-out-loud and spoken tasks (i.e., verbal fluency difficulties) in early stages of AD [57]. These speech characteristic changes may translate through to keystroke dynamics as well. Researchers using passive data can measure the frequency of text messages, the duration for text messages to be typed, and other keystroke dynamics to infer these changes. In one study, symptomatic participants with MCI or AD received less text messages and sent less text messages than healthy controls [46]. Additionally, these symptomatic participants with MCI or AD had slower and more variable typing and tracing outcomes in different tasks on an assessment application. Another study, using an application which replaces the built-in keyboard with the application’s own custom keyboard to collect passive data, asked participants to complete structured assignments [47]. These assignments were to type paragraph-length texts as a response to a prompt on their smartphones. These assignments were performed in a non-clinical setting without autocorrect or a time limit, and then participants were asked to send these texts to the researchers so that they could be analyzed. Researchers found that participants with MCI used less nouns than verbs in the structured assignment. Additionally, using six months of passively acquired keystroke data along with natural language processing, researchers were able to detect mild cognitive impairment in patients and distinguish them from controls [47]. By discerning these subtle changes in texting, smartphones provide a potential way to detect MCIs and monitor cognitive fluctuations, allowing for treatments or close monitoring to be implemented earlier to improve the quality of life for patients with neurodegenerative disorders. 

### 3.2. Multiple Sclerosis

Multiple sclerosis (MS) is a neurodegenerative immune-mediated disorder causing mobility and cognitive impairment as immune cells attack neurons in the central nervous system [58,59]. These impairments can be present early in the disease course, and atrophy captured by MRI can also be seen early on in the disease course [60]. Given that cognitive impairment is evident early on, being able to detect MS at its onset or near after provides a crucial window to stem the further progression of the disease. Thus far, studies have used smartphone applications (e.g., elevateMS) to assess motor and cognitive functions in patients with MS [61], but were impeded by incomplete data assessments that required active participation from patients in order to monitor symptoms and disease burden. Using passive data allows researchers to obtain data longitudinally, near-continuously, and unobtrusively, thus bypassing these obstacles. Indeed, by using longitudinal keystroke dynamics, researchers have been able to extract potential biomarkers for multiple sclerosis [49]. In one study, typing sessions were initially aggregated per day to obtain five summary statistics: mean, median, standard deviation, minimum, and maximum. Patients with MS had on average significantly higher keystroke latencies compared to controls. These keypress latencies were positively correlated with the expanded disability status scale (EDSS), while key release was positively correlated with the nine-hole peg test (NHPT). All keystroke features were negatively correlated with the symbol digit modalities test (SDMT). The median time of disease duration in patients was 5.7 years and the median of disease severity, using the EDSS, was 3.5 years within this cohort. Even with mild disease severity and with a shorter disease duration in patients with MS, distinctions between controls and patients were already apparent. Another study also examined the relationship between keystroke dynamics and cognitive functioning in participants with MS [48]. They found that typing speed and use of the backspace key along with autocorrection events correlated with a better cognitive functioning and less severe symptoms. These correlations imply that participants with MS who have more mild symptoms could potentially be better at monitoring and correcting their mistakes. Another study was able to group participants by detecting bradykinesia and rigidity in users’ dominant hands using machine learning algorithms on keystroke features [50]. Using one year’s worth of data, researchers found that participants with MS who had worse arm motor function had a higher latency between keypresses, and participants with MS who had a decreased processing speed corresponded with a higher latency using punctuation and backspace keys [50]. Using the same dataset, researchers were also able to estimate the levels of disease severity, manual dexterity, and cognitive capabilities from keystroke dynamics using a machine learning model that used three predictors (a time-related cluster, a cognitive-related cluster, and the number of times autofill was used) [51]. Participants with MS who were quicker to correct and adjust their texting had higher SDMT scores, an indicator of cognitive functioning, which helped with model predictions [51]. These studies show that keystroke dynamics can be used as potential biomarkers for MS before significant disease onset, which would allow for earlier treatments and preventative care. 

## 4. Keystroke Dynamics and Affected Cognitive Domains in Mood Disorders

Certain mood disorders are associated with cognitive deficits [62,63,64], with cognitive deficits being established through neuropsychological tests for bipolar disorder [65,66,67] and depression [63,68,69]. Cognitive deficits that can be found in patients with mood disorders imply a disruption in cognitive control [70,71]. Cognitive control is a necessary ability to flexibly alter and guide behavior in the face of constantly changing circumstances, which is hindered in those with mood disorders. To examine cognitive control, task-switching paradigms (i.e., trail-making test part B) test cognitive flexibility [72], processing speed [73], and executive control [74]. Previously, these tests were administered in person via pencil and paper but have now been adapted and validated for digital devices (i.e., smartphones) [45,75]. Recently, researchers used smartphones and passive data collection to examine cognitive control in participants with mood disorders [52]. They found that participants with mood disorders not only showed lower cognitive performances on the trail-making test part B, but participants with mood disorders also had diurnal pattern differences in their keystroke dynamics compared to healthy controls, where individuals with higher cognitive performances had faster keystrokes and more consistent typing speeds throughout the day [52]. Another study examined processing speed and executive function in patients with bipolar disorder by comparing keystroke dynamics with a smartphone-based version of a task-switching paradigm and a depression rating scale [53]. Researchers found that typing speeds from keystroke dynamics, especially when compared to mood ratings, could potentially derive features of cognition and cognitive control, such as visual attention, processing speed, and task switching. 

Changes in linguistic patterns can reflect certain mood states [76], and smartphones can provide a way to potentially measure these changes in mood in a non-clinical setting as well as provide objective measurements. Previously, patients with bipolar disorder in a depressive state were shown to have an impairment in phonemic fluency, while patients with bipolar disorder in a manic state were shown to have a moderate-to-large effect size deficient in language when it came to letter fluency and semantic fluency [66]. Recently, using smartphones and passive data collection, researchers examined keystroke dynamics and found that participants with bipolar disorder who had more depressive symptoms had increased autocorrect rates, while participants with bipolar disorder who were in a potentially more manic state used the backspace key less [54]. This can possibly be accounted for by a decreased ability to concentrate within depressed states and additionally a decreased self-monitoring known to happen with higher mania scores. In another study, researchers investigating keystroke dynamics in patients with depression found that patients with depression had longer hold times between both pressing and releasing a key and between releasing a key and pressing the next one [55]. Distilling these subtle changes in keystroke dynamics, especially in conjunction with depression scores, would allow researchers and clinicians to monitor any potential cognitive dysfunction, which would allow for early intervention or treatment for particular mood disorders. Early intervention could be crucial and provide life-saving treatment.

## 5. Discussion

The aim of this review was to examine how researchers have been using keystroke dynamics from smartphones to examine cognition. Keystroke dynamics have provided potential digital biomarkers to infer cognitive functioning outside of the laboratory. In general, smartphones allow for unobtrusive, near-continuous, and longitudinal passive data collection, which provides a unique means for future research directions. From having individual cognitive footprints [77] to predicting mood states [54], keystroke dynamics coupled with accelerometry appears to provide sufficient informative data to distinguish healthy controls from people with neuropsychiatric disorders. Currently, most studies implementing passive data to investigate cognition have been through a lens of neuropsychiatric disorders, extracting potential biomarkers from keystroke dynamics and other passive data. Although these biomarkers show promising results, more research must be conducted before remote diagnoses or disease monitoring can replace expert evaluation. Additionally, as passive data collection and analyses become more advanced, perhaps the lens examining disorders can be expanded, and the biomarkers found from examining disorders can be applied universally for preventative measures and early diagnoses. However, there currently remain many obstacles to assessing neurocognitive functioning and predicting cognitive fluctuations. One limitation we encountered was that some studies we reviewed found potential biomarkers through detecting statistically significant group differences in keystroke dynamics, while other studies used predictive models utilizing features from keystroke dynamics. Studies that used only statistical analyses could apply prospective biomarkers to predictive models and examine the capability of said biomarkers. Another limitation we encountered for studies using statistical analyses that specifically used mixed-effect models were that there were no effect sizes for these analyses, given the complexity of different variances at each level. Additionally, sample sizes were often small or skewed toward those with disorders. We would suggest implementing a longitudinal follow-up of cohorts consisting of those with neuropsychiatric disorders to observe if the same digital passive biomarkers can indeed predict cognition in subsequent in-person evaluations. Another potential biomarker from passive data would be using GPS location entropy, a measure of regularity, in conjunction with keystroke dynamics. These could potentially yield new predictive biomarkers to provide warnings before significant mood or cognitive changes, as decreased entropy in GPS location data have previously been associated with depression [78,79,80]. 

As previously mentioned, cognition, as defined by RDoC, has six domains [16], but not all were quantifiable. Some domains were not elaborated on due to a current lack of research comparing passive data measurements to the gold-standard measurements of these domains. This is a gap in the literature that can be improved upon and is an essential area to investigate, given the negative implications of declining attention, perception, working, and declarative memory. With innovative large language models being created and updated, there may soon be ways to combine them with keyboard dynamics to examine cognition and further research toward specific cognitive domains. 

Ethical concerns certainly arise regarding informed consent, smartphone usage, passive data collection, and privacy. Data literacy can be a key component of informed consent for certain research studies. For some neuropsychiatric disorders with longitudinal studies, obtaining participant consent multiple times throughout the study may be necessary and would be ethically important as the disease course progresses. As some neuropsychiatric disorders progress, participants may reconsider their participation in research studies and may want to opt-out of studies or may even lack the cognitive facilities to make informed choices. Because of the dynamic nature of cognition for certain neuropsychiatric disorders, researchers must be cognizant of these possibilities and may want to periodically confirm participant consent, potentially using over-the-air updates via smartphones for this purpose. Researchers also need to ensure that participants not only understand the purpose for the research being conducted but researchers also need to be able to explain the technology being used in layman terms to ensure full comprehension. 

In terms of passive data collection and privacy, with the enormous amounts of data being continually collected from participants, researchers not only need to ensure that participants understand the extent of data collection, but are also obligated to handle the data securely and to anonymize the data when necessary. In 2018, the European Union (EU) has passed the General Data Protection Regulation (GDPR) law to protect the right to privacy for its residents [81]. The GDPR strengthens data rights of all EU residents and holds data controllers accountable to keep digital data private. In comparison, the USA does not currently have an equivalent comprehensive law that protects consumer data. There are niche laws that protect certain types of personal data (e.g., Health Insurance Portability and Accountability Act (HIPPA), Electronic Communications Privacy Act (EPCA), Children’s Online Privacy Protection Rule (COPPA), Family Educational Rights and Privacy Act (FERPA), etc.) that currently provide some protection for digital privacy. However, there are several comprehensive acts of legislation for data privacy, at both the state and federal levels, that are underway and meant to protect individual data privacy. As passive data collection becomes more utilized, perhaps government oversight (e.g., the GDPR law) is necessary to have a universally accepted method of storing and encrypting data to ensure the privacy of participants. 

In the realm of digital privacy, all smartphone users emit digital exhaust. Digital exhaust encompasses all the information that smartphone users create and leave behind as they browse websites and applications. By data mining digital exhaust, researchers could study individuals and their unique patterns and potentially distinguish individuals from their data. Researchers could then derive more information regarding specific individuals, jeopardizing their right to privacy, especially in a medical context, as being able to identify individuals could be harmful if the data were breached and then searchable. As increased amounts of data can be extracted from smartphone usage, the distinction between personal digital data and health data becomes more obscure, as evidenced by the studies reviewed herein that demonstrated the utility of keystroke dynamics in passively inferring cognition. It then becomes the responsibility of researchers to ensure the privacy of their research participants. With participants’ privacy in mind, researchers could examine smartphone users’ digital exhaust to explore cognitive domains. However, with its dynamic nature, the varying amounts of digital exhaust can modulate users’ cognitive performance as well. 

Arguably, the relationship between smartphone usage and cognition is a complicated one, with research suggesting not only negative effects but also positive and potentially neutral ones. Smartphone usage, especially in cases of excessive use and in addiction, has garnered a negative reputation by being a distraction from tasks at hand [82] and potentially decreasing concentration levels and worsening impulse control [83]. Excessive smartphone usage is also potentially associated with higher rates of depression, anxiety, and smartphone addiction [84,85], which can negatively affect cognition and lower academic performance [86]. Additionally, smartphone users may perceive their smartphone usage as less than their actual usage [5,87,88]. Furthermore, smartphone notifications can be distractive by provoking phone checking, which might lead to habitual checking [89,90] and, potentially, smartphone addiction [91]. This in turn can interrupt attention and focus on other tasks [92,93], decreasing executive function [94] and increasing cognitive failures [95]. Moreover, social media applications on smartphones can also have a negative impact on cognitive control [96], especially in adolescents [97], and an increased use of these applications can lead to cognitive failures [95,98], along with being a risk factor for worsening mental health [99,100]. In addition, available cognitive capacity can decline when in the mere presence of a smartphone [101], and having a smartphone within view can cause distractions from the task at hand and impair productivity, despite not being actively checked [92,102]. In contrast to these negative findings, some studies could not find strong evidence demonstrating the detrimental effects of smartphones on cognition [103,104]. Furthermore, some research has also shown that smartphones may aid cognition and memory through various means. Some smartphone applications may help in decreasing cognitive load through certain tools and applications (i.e., writing down tasks in a list using a smartphone, smartphone calendar reminders for appointments, registering contacts’ phone numbers) [95], allowing for said cognitive space to be used for other purposes. In one study, participants were able to more accurately complete a task by using smartphones to help record and remember parts of the task [105], but when smartphones were taken away in a subsequent task, participants fared worse than when they had not depended on a smartphone originally. Smartphones can also have specific applications geared toward training better cognitive function effectively, especially for the elderly [106]. As digital health technologies become more widespread, researchers using passive data collection to study cognition should be cognizant of this complex relationship and find ways to bypass any incongruencies. 

In conclusion, for the last decade, smartphones have provided researchers with a new device and avenue for health technologies. As smartphones become even more ubiquitous, their impact on health research exponentially increases as well. The number of features that researchers have been able to extract thus far from passive data collected from digital technologies already have numerous clinical applications. In this review, we have shown the utility of using keystroke dynamics and the richness it can provide for data analyses, especially when compared in conjunction with other passively and actively collected data. However, there remains innumerous features that can be extracted further from passive data, and ensuring digital data privacy for participants and obtaining their consent for longitudinal studies, especially for those with neuropsychiatric and neurodegenerative disorders, is key.

## Figures and Tables

**Figure 1 brainsci-13-00959-f001:**
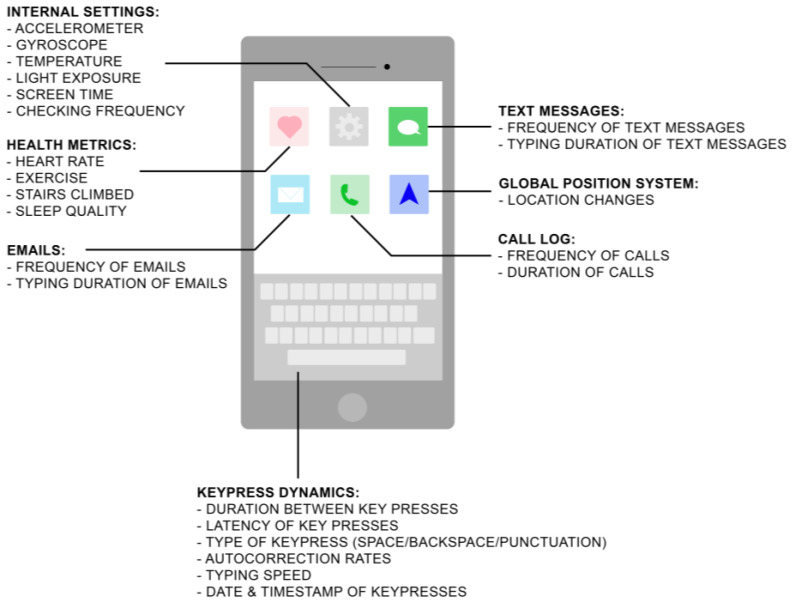
Smartphone schematic depicting examples of passive data collection.

**Figure 2 brainsci-13-00959-f002:**
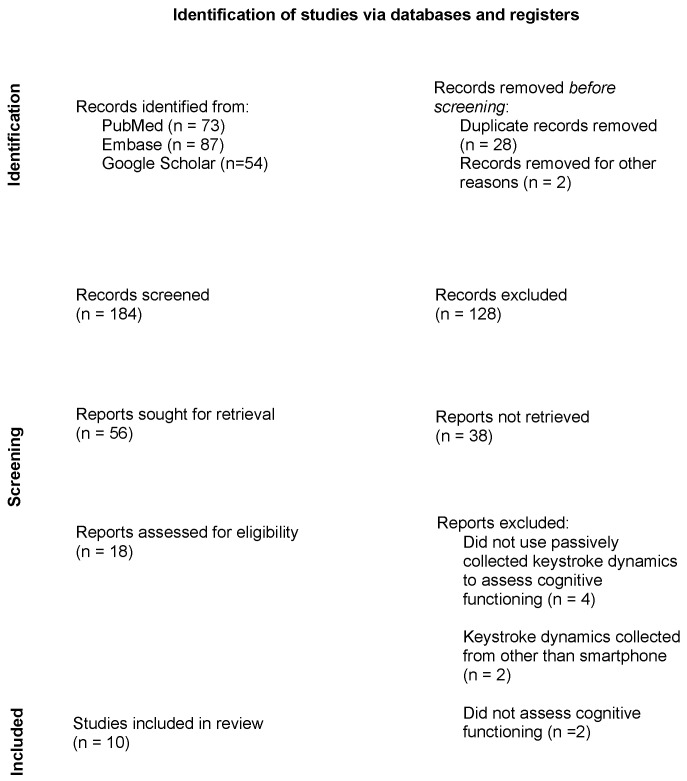
PRISMA flow diagram of search and selection process.

**Table 1 brainsci-13-00959-t001:** Definitions of passive and active data collection along with examples.

Type	Definition	Examples	Implementable Digital Devices
Active Data Collection	Data acquisition from participants requiring active participation, allowing for subjective data measurements	Ecological momentary assessments, mood/cognitive/motor self-reported assessments	Smartphones, tablets, smartwatches
Passive Data Collection	Unobtrusive data acquisition from participants from digital technology, where participants are unaware of collection, allowing for objective data measurements	Keystroke dynamics, accelerometer, GPS, screen time, temperature, phone-checking frequency, physical activity, number of text messages and emails, duration and frequency of phone calls made	Smartphones, tablets, smartwatches, sleep monitors, fitness trackers

**Table 2 brainsci-13-00959-t002:** Summary of study characteristics and findings.

Study	*N* Participants	Digital Technologies Implemented	Data Collected	Findings	Statistical Analysis	Machine Learning Models and Validation Metrics
Chen, R. et al., 2019 [46]	24 people with mild cognitive impairment (MCI), 7 people with mild AD dementia, 84 healthy controls	iPhone 7 Plus, Apple Watch Series 2, 10.5” iPad Pro with a smart keyboard, Beddit sleep monitoring device	Passive: Number of text messages sent and received, time duration to send a text message, typing speeds, accelerometer, gyroscope, stairs climbed, stand hours, workout sessions, heart rate, sleep sensors, application usage time, phone unlocks, breathe sessions Active: Daily energy surveys, tapping task, dragging task, typed narrative task, verbal narrative task, video, and audio bi-weekly	Symptomatic participants (with MCI or with mild AD dementia) typed slower, had a less regular routine (measured via first and last phone acceleration), took their first steps of the day later (measured via phone’s pedometer), sent and received fewer text messages, relied more on applications suggested by Siri, and had worse survey compliance than healthy controls	Not Applicable (N/A)	Model: Extreme gradient boosting algorithm Validation: training/testing: 70/30 Results: demographics AUROC: 0.757; device-derived features AUROC: 0.771 (±0.016, 95% CI); demographics + device derived features = 0.804 (±0.015, 95% CI); age-matched demographics AUROC: 0.519 (±0.018, 95% CI); age-matched device features AUROC: 0.726 (±0.021, 95% CI); age-matched demographics + device features AUROC: 0.725 (±0.022, 95% CI)
Ntracha, A. et al., 2020 [47]	11 people with MCI, 12 healthy controls	Android smartphones	Passive: Keystroke dynamics (timestamps of keypresses and releases, backspace, pauses, number of characters typed, typing session duration) Active: PHQ-9 Questionnaire, written assignments for natural language processing (typing up to four paragraphs on a given topic)	Participants with mild cognitive impairment (MCI) were able to be distinguished from healthy controls using passive and active data in natural learning processing models, keystroke models, and fused models. Participants with MCI had bradykinesia and rigidity detected from their keystroke dynamics when compared to healthy controls	N/A	Models: k-Nearest Neighbors (k-NN), logistical regression (LR), random forest, ensemble method Validation: leave one subject out method for training and testing Results: Keystroke features with kNN classifier: AUC: 0.78 (0.68–0.88, 95% CI), specificity/sensitivity: 0.64/0.92 Natural Language Processing (NLP) features with LR classifier: AUC: 0.76 (0.65–0.85, 95% CI), specificity/sensitivity: 0.80/0.71 Ensemble model fusion of keystroke and NLP features: AUC: 0.75 (0.63–0.86, 95% CI), specificity/sensitivity: 0.90/0.60
Chen, M. et al., 2022 [48]	16 people with multiple sclerosis (MS), 10 healthy controls	iOS and Android smartphones	Passive: Keystroke dynamics (keypress type, timestamp, relative distance between consecutive keystrokes, distance between the keystroke and the center of the keyboard), accelerometer Active: Digital neuropsychological tests (symbol digit modalities test, digit span, trail-making test, Delis–Kaplan executive function system (D-KEFS) color-word interference test, controlled oral word association test or D-KEFS verbal fluency test, California verbal learning test, Rey auditory verbal learning test, or Hopkins verbal learning test-revised), symptom rating scales (modified fatigue impact scale, Chicago multiscale depression inventory, state-trait anxiety inventory)	Participants with MS with less severe symptoms had higher uses of the backspace key and a faster typing speed. Faster typing speed was associated with better performance on measures of processing speed, attention, and executive functioning as well as having less impact from fatigue and having less severe anxiety symptoms	Method: Multilevel models (level 1: keystroke dynamics within typing session; level 2: subjects) Significant results: Features evaluated using Welch’s *t*-test: number of days of data collection (mean number): −1.86, *p* = 0.076 proportion of time spent using one hand to type (%): 542.70, *p* < 0.001 number of characters per typing session (mean): 0.01, *p* < 0.107 median inter-key delay (typing speed) per session (seconds): −1.45, *p* < 0.001 inter-key delay median absolute deviation per session (seconds): 0.11, *p* = 0.032	N/A
Lam, K.H. et al., 2020 [49]	102 people with MS, 24 healthy controls	iOS and Android smartphones	Passive: Keypress dynamics (type of keypress (alphanumeric, backspace, space key, punctuation), time and date of keypresses, successive keypress latencies and releases Active: Assessments, including expanded disability status scale, nine-hole peg test, symbol digit modalities test (SDMT)	Participants with MS had higher keypress latencies, release latencies, flight time, post-punctuation pause, pre-correction and post-correction slowing compared to healthy controls	Method: Pearson’s correlation coefficient Significant results: SDMT with: press-press latency: −0.525, *p* > 0.01 release-release latency −0.553, *p* < 0.01 hold time: −0.286, *p* < 0.01 flight time: −0.525, *p* < 0.01 pre-correction slowing: −0.300, *p* < 0.01 post-correction slowing −0.444, *p* < 0.01 correction duration: −0.162, *p* < 0.05 after punctuation pause: −0.317, *p* < 0.01	N/A
Lam, K.H. et al., 2022 [50]	102 people with MS	iOS and Android smartphones	Passive: Keypress dynamics (type of keypress (alphanumeric, backspace, space key, punctuation), time and date of keypresses, successive keypress latencies and releases Active: Assessments, including expanded disability status scale, nine-hole peg test, symbol digit modalities test (SDMT)	Participants with MS with worse arm function had higher latency between keypresses and participants with worse processing speed corresponded with higher latency using punctuation and backspace keys	Method: Linear mixed-models Significant results: cognitive score cluster associated with SDMT: −8.57 (−12.02 to −5.12, 95% CI), *p* < 0.001, random effect variance: 82.7%, explained variance: 25.4%; cognitive score cluster and covariances (age, sex, level of education): −5.02 (−9.02 to −1.02), *p* = 0.02, random effect variance: 77.1%, explained variance: 30.4%; hybrid model (including covariates): between subjects: −11.25 (−17.28 to −5.21), *p* < 0.001; within subjects: −0.35 (−5.60 to 4.89), *p* = 0.9	N/A
Hoeijmakers, A. et al., 2023 [51]	102 people with multiple sclerosis (MS), 24 healthy controls	iOS and Android smartphones	Passive: Keypress dynamics (type of keypress (alphanumeric, backspace, space key, punctuation), time and date of keypresses, successive keypress latencies and releases Active: Assessments, including expanded disability status scale, nine-hole peg test, symbol digit modalities test	Participants with MS could be discerned from healthy controls by using clinical outcome measures as targets for machine learning (ML) techniques, with ML techniques being able to estimate level of disease severity, manual dexterity, and cognitive capabilities	N/A	Models: Binary classifications: random forest, logistical regression, k-nearest neighbors, support vector machine, Gaussian naive Bayes Validation: training/testing: 80/20 Results from cross-validation: AUC = 0.762 (0.677–0.828, 95% CI) AUC-ROC = 0.726, sensitivity/specificity/accuracy: 0.750/0.429/0.48 estimating level of fine motor skills AUROC: 0.753
Ning, E. et al., 2023 [52]	64 participants with mood disorders (major depressive disorder, bipolar I/II, persistent depressive disorder, or cyclothymia), 26 healthy controls	iOS and Android smartphones	Passive: Keystroke dynamics (category of keypress (i.e., alphanumeric, backspace, punctuation), associated timestamps, autocorrection events), accelerometer, gyroscope Active: Digital trail-making tests part B	Participants with mood disorders showed lower cognitive performance on the trail-making test. There were also diurnal pattern differences between participant with mood disorders and healthy controls, where individuals with higher cognitive performances had faster keypresses and were less sensitive to the time of day	Method: longitudinal mixed effects Significant results: aging effect: typing slowed ~20 ms/7 years; sessions with lower accuracy had shorter IKDs ~10 ms, *b* = −0.89, *p* < 0.001; more variable IKD within a session has slower session typing, *b* = 434.57, *p* < 0.001; more typing, faster typing, *b =* −4.35, *p* < 0.001	N/A
Ross, M. et al., 2021 [53]	11 people with BP, 8 healthy controls	Samsung Galaxy Note 4 Android smartphone	Passive: Keystroke dynamics (category of keypress (i.e., alphanumeric, backspace, punctuation), associated timestamps, autocorrection events) Active: Digital trail-making tests part B (dTMT-B)	Participants with mood disorders had significantly different keystroke dynamics from healthy controls when compared to depression ratings and the trail-making test	Method: longitudinal mixed effects Significant results: subject-centered HDRS-17 score predicting dTMT-B: *b* = 0.038, *p* = 0.004; subject-centered typing speed predicting dTMT-B: *b* = 0.032, *p* = 0.004; faster grand mean centered typing speed suggesting faster dTMT-B completion time: *b* = 0.189, *p* < 0.001	N/A
Zulueta, J. et al., 2018 [54]	9 people with bipolar disorder (BP) (5 with BP I, 4 with BP II)	Samsung Galaxy Note 4 Android smartphones	Passive: Keystroke dynamics (keystroke entry date and time, duration of keypress, latency between keypresses, distance from last key along two axes, and autocorrection, backspace, space switching-keyboard, and other behaviors), accelerometer Active: Hamilton depression rating scale, Young mania rating scale	Participants with bipolar disorder who were in a potentially more manic state (had higher mania symptoms) used the backspace key less and while in a potentially more depressive state had an increase in autocorrection rates	Method: mixed effects regression Significant results: average accelerometer displacement with HDRS: 3.20 (1.20 to 5.21, 95% CI), *p* = 0.0017; average accelerometer displacement with YMRS: 0.39 (0.15 to 0.64, 95% CI) *p* = 0.003; autocorrect rate with HDRS: 2.67 (0.87 to 4.47, 95% CI), *p* = 0.0036; backspace ratio with YMRS −0.30 (−0.53 to −0.070, 95% CI), *p* = 0.014	N/A
Mastoras, R.E. et al., 2019 [55]	11 people with depressive tendencies, 14 healthy controls	Android smartphones	Passive: Keystroke dynamics (timestamps of keypresses and releases, delete rate, number of characters typed and typing session duration) Active: Patient Health Questionnaire-9	Participants with depressive tendencies held down keypresses for longer and had longer pauses between keypresses compared to healthy controls	N/A	Models: Random forest, gradient boosting classifier, support vector machine classifier Validation: leave one subject out method for training and testing Results: random forest (best performing pipeline): AUC = 0.89 (0.72–1.00, 95% CI), sensitivity/specificity: 0.82/0.86

## Data Availability

Not applicable.

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
