# Peer review of "A Review on Smartphone Keystroke Dynamics as a Digital Biomarker for Understanding Neurocognitive Functioning"

_brainsci, 2023, doi:10.3390/brainsci13060959_

Round 1
Reviewer 1 Report (Previous Reviewer 2)
My concerns in the previous review have been addressed well. There is a significant improvement in the current version of the manuscript.
Author Response
We thank the reviewer for their time.
Reviewer 2 Report (New Reviewer)
The paper presents a review of studies that have analyzed passively-collected keystroke data from smartphones to distinguish between patients with cognitive impairment or mood disorders and healthy controls. It also provides a useful introduction to the what kinds of data can be collected from smartphones, balancing a discussion of clinical and scientific upsides with real concerns about privacy. The authors conclude that keystroke data is a rich source of information that may yield important clinical insights with higher temporal resolution than traditional screening techniques that require active data collection.
I appreciate how thoroughly the authors address the ethical concerns involving using data of this kind--including those involving "digital exhaust" that everyone on the internet is leaving in their wake. I agree with the authors that this is a sensitive topic, and even if there is great promise in mining keystroke data we need to consider the consequences of developing tools to gain insight into a person's cognitive state without consent.
My primary concern is that the paper has a clinical outlook (i.e., can keystroke data be used to detect MCI or predict the onset of dementia?), but the results are presented in terms of significant group differences. Detecting a difference between group may be sufficient for testing a specific hypothesis, but may not correspond with the ability of a model to reliably predict whether an unlabeled case is a patient or control. This distinction is discussed in many papers on clinical predictive modeling; one paper that addresses the limits of data-driven modeling for suicide prediction raises a number of important concerns https://doi.org/10.1016/j.cpr.2020.101940. One concern is that most people do not have a disorder; models that seem to work when the groups are balanced or which over represent the clinical group (as in most of the work you have reviewed that mines keystroke data). Another is that building data-driven models that are not constrained by theory may be hard to interpret and have limited generalizability.
To begin their discussion, the authors state that "The aim of this review was to examine how researchers have been using keystroke dynamics from smartphones to examine cognition". One can examine cognition to test neurocognitive or clinical hypotheses and/or to discern predictive "biomarkers". The review seems to emphasize the search for biomarkers, which is fine, but I think the distinction should be more clear when you are talking about one thing vs. the other, because the kind of statistical evidence required differs between them.
To be more concrete about what I think should be done:
1. It should be made clear whether group differences were assessed with something like an ANOVA or t-test, or with a predictive model.
2. If a predictive model was used, it should be noted whether performance was assessed using cross-validation or a separate validation sample.
3. Effect sizes from the studies reviewed should be reported. For predictive models, this might be sensitivity/specificity.
4. The distinction between statistically significant differences in experiments vs. predictive models should be emphasized, in light of the review's focus on discovering biomarkers.
5. The challenges of building predictive models/discovering generalizable biomarkers atheoretically and/or in light of the discrepant prevalence of disordered vs. healthy individuals in the populations should be acknowledged.
In short, I feel like the primary weakness of the review is that it seems to conflate statistical significance with the ability to make generalizable predictions of future unlabeled cases by not addressing this distinction explicitly. This is exacerbated by not reporting on how the work reviewed arrived at their effects and by not reporting any stats or effect sizes that quantify these differences. A major contribution of this review could be providing an overview of observed effect sizes with varying passive metrics and clinical groups.
My secondary concern is that the methods section is very terse. It is typical to report how many papers were returned by the initial search, and how many were excluded/retained after each step in the decision protocol. Many systematic reviews diagram out their decision protocol and report those numbers within the diagram. It is also necessary to report whether the search was performed over the title, abstract, and/or content of the document. The method section must be expanded to include all details necessary to replicate.
Minor concerns
1. The paragraph beginning on line 372 is very interesting as a stand-alone piece of writing, but it felt like a digression. The review is not about whether smartphone use is a net positive or negative; it is acknowledging that smartphones are widely adopted and extensively used, and this presents important opportunities.
2. There are points throughout the paper where the authors state the obvious before getting to their point. For example, on line 133: "Language is an important tool necessary to communicate and connect to others in society". This can be taken for granted in the interest of tightening up the prose.
3. Thereafter should be hereafter on line 97
4. "were" should be "are" on line 182.
5. Review the reference section: reference 7 is to a book review, rather than the textbook itself,
Author Response
Please see the attachment.

This manuscript is a resubmission of an earlier submission. The following is a list of the peer review reports and author responses from that submission.
Round 1
Reviewer 1 Report
Thank you so much for the opportunity to review this manuscript. The topic is important and looks interesting. However, the process of selection of the manuscript in this review is unclear. Other major concerns are as follows:
1. L128. L134.
The authors mentioned that this review will focus on passive data collected by smartphone. Yet, in many parts of the manuscript, active data collection and passive data collection were mixed. Can the author clarify the definition of active vs passive data collection and organize the manuscript flow?
2. L145
Selecting the topic and studies related to smartphone addiction in the context of attention is a bit surprising since the addition won't be in line with solely attention context.
3. L 166
Similar to comment 2, it is out of the blue to introduce stress here.
4. L208
Why did the author introduce perception here? The logical flow of this review is hard to follow. I won't comment on all of the similar issues here. Language and Alzheimer etc. Most of the topics they selected did not logically flow very well.
5. The authors mentioned that the aim of this review is to investigate how researchers used passive data collected by smartphones to examine cognition. The aim is too broad to provide a concise and useful review for readers. How did they define "Cognition" here? Why does it have to be smartphone data? How did they select the area of topic and relevant studies were not clear.
Reviewer 2 Report
This is an interesting review on passive unobtrusive data collection from digital technologies to measure attention, perception, language, and cognitive control. The paper is timely and I agree that it may have a good contribution to the literature. I have several major comments to improve the manuscript further:
1. First, I am not convinced that measurement of smartphone addiction should be categorized under attention. It is also unclear the research that is reviewed by the authors are really measuring smartphone addiction or simply smartphone usage. The link between smartphone addiction and attention is also unclear.
2. Although there is a link between stress and attention, I don't think that measurement of stress is a good proxy of attention. Both constructs are theoretically distinct even though they may affect each other. If stress is a good proxy of attention, why it is also not a good proxy of cognitive control and perception too? It will be important for the authors to support their argument further. I would suggest the authors to consider to argue stress under perception.
3. Under attention (or cognitive control), I think the authors should consider to review literature on the relationship between smartphone checking and cognitive failure. That might be more proximal and relevant than stress under the category of attention.
4. I think the current subcategory (e.g., mood disorders, smartphone addiction) under each cognitive domains are difficult to follow. It will be important for the authors to specify it clearer
5. I would suggest the authors to summarize each domain, smartphone data, and outcomes in a comprehensive table to help the readers to understand the paper easier.
6. In the discussion, the authors argue that smartphone can negatively affect cognition. To provide a more balance review, recent studies have also showed that some type of smartphone use may benefit cognitive functions too. The authors should also highlight these recent findings.
Relevant paper:
Smartphone use and daily cognitive failures: A critical examination using a daily diary approach with objective smartphone measures. British Journal of Psychology. https://doi.org/10.1111/bjop.12597